# Molecular engineering of safe and efficacious oral basal insulin

Frantisek Hubálek [1], Hanne H. F. Refsgaard[1], Sanne Gram-Nielsen [1], Peter Madsen [1], Erica Nishimura [1], Martin Münzel [1], Christian Lehn Brand[1], Carsten Enggaard Stidsen [1], Christian Hove Claussen[1], Erik Max Wulff [1], Lone Pridal[1], Ulla Ribel[1], Jonas Kildegaard[1], Trine Porsgaard[1], Eva Johansson [1], Dorte Bjerre Steensgaard[1], Lars Hovgaard[1], Tine Glendorf[1], Bo Falck Hansen[1], Maja Kirkegaard Jensen[1], Peter Kresten Nielsen[1], Svend Ludvigsen[1], Susanne Rugh[1], Patrick W. Garibay[1], Mary Courtney Moore[2], Alan D. Cherrington [2] & Thomas Kjeldsen [1]✉

Recently, the clinical proof of concept for the first ultra-long oral insulin was reported, showing efficacy and safety similar to subcutaneously administered insulin glargine. Here, we report the molecular engineering as well as biological and pharmacological properties of these insulin analogues. Molecules were designed to have ultra-long pharmacokinetic profile to minimize variability in plasma exposure. Elimination plasma half-life of ~20 h in dogs and ~70 h in man is achieved by a strong albumin binding, and by lowering the insulin receptor affinity 500-fold to slow down receptor mediated clearance. These insulin analogues still stimulate efficient glucose disposal in rats, pigs and dogs during constant intravenous infusion and euglycemic clamp conditions. The albumin binding facilitates initial high plasma exposure with a concomitant delay in distribution to peripheral tissues. This slow appearance in the periphery mediates an early transient hepato-centric insulin action and blunts hypoglycaemia in dogs in response to overdosing.

[1] Novo Nordisk A/S, Novo Nordisk Park 1, 2760 Maaloev, Denmark. [2] Vanderbilt University, Nashville, TN, USA. ✉email: ThK@novonordisk.com

The availability of oral insulin would free millions of people with diabetes from the everyday burden of subcutaneous (SC) insulin injections. Furthermore, the convenience of oral insulin might facilitate earlier initiation of insulin therapy and better patient compliance leading to improved glucose control and long-term outcomes. Two key challenges need to be overcome to develop an oral insulin therapy: (1) sufficient bioavailability and (2) clinical performance including efficacy, safety, and especially low pharmacodynamic variation. Pharmacodynamic variability is particularly challenging for insulin, due to the large absorption variability associated with oral dosing in combination with a narrow therapeutic window with an overdose being associated with potentially life-threatening hypoglycaemia.

The first attempt to deliver insulin orally was conducted soon after the discovery of insulin nearly 100 years ago[1] and it was concluded that the variability associated with oral administration of insulin would render it of little or no therapeutic value. Nevertheless, over the years, numerous technologies have been explored, including, e.g., nanoparticles, hydrogels, liposomes, ionic liquids or ingestible devices[2–6]. Despite relatively high bioavailability reported in several rodent studies[7], most of these attempts have not progressed past research or the preclinical stage and consequently there is currently no oral insulin therapy available[5]. This can at least partly be ascribed to the use of human insulin or rapid-acting insulin analogues[8]. In contrast, we rationalised that ultra-long acting basal insulin analogues can overcome this limitation by addressing the critical issue of variability due to accumulation to steady state combined with a protracted action profile. Thus, an elimination half-life of >24 h combined with once daily dosing will lead to a steady-state exposure level accumulated from the dose of multiple consecutive days as illustrated in Fig. 1.

Here we describe the engineering of insulin analogues for safe, efficacious, oral basal insulin therapy in man[9] with no impact of food intake when the tablet is administrated 30 min before a meal[10] using Gastro Intestinal Permeation Enhancer Tablet I (GIPET®I)[11] based on sodium caprate as absorption enhancer. The combination of enhanced proteolytic stability with ultra-low receptor affinity and strong albumin binding leads to significantly improved bioavailability and ultra-long plasma half-life, while maintaining potency with regards to whole-body glucose disposal under euglycemic clamp conditions. Furthermore, we demonstrate that the inherent properties of the molecule may curtail the hypoglycaemic response to overdosing in normal dogs. Importantly, these effects are linked to the peaked absorption profile, observed both in dog and man[9] which can be achieved only by oral, and not by SC administration.

## Results

**Design of proteolytically stable ultra-long insulin.** First, we aimed at reducing the proteolytic degradation of insulin to improve the bioavailability after oral dosing. Initially, a large number of insulin analogues with substitutions at known enzymatic (primarily trypsin/chymotrypsin) cleavage sites[12] were produced and screened for their rate of proteolytic degradation. Two amino acid substitutions (A14 Tyr→Glu and B25 Phe→His, named OI106) were sufficient to stabilize the insulin molecule, with an optional additional deletion of B27Thr (named OI189) conferring further stability (Fig. 2b). Concomitantly, the substitutions lowered the relative insulin receptor affinity by threefold, while considerably increasing solubility in the slightly acidic pH range (5.5–6.5) found in the small intestine[13] (Supplementary Fig. 1).

Secondly, mechanisms to prolong the pharmacokinetic profile of the insulin were incorporated into the molecule. Currently available long-acting injectable insulin analogues are mostly prolonged by slowing the rate of absorption from the injection site into the circulation (e.g. insulin detemir, insulin glargine and insulin degludec)[14]. However, for oral insulin delivery, it is critical that the full daily dose is rapidly absorbed to avoid interaction with ingested food. Therefore, we designed insulin analogues which were protracted in the circulation with an elimination half-life of >24 h. We rationalized that this could be achieved mainly by slowing down clearance, primarily by two mechanisms: (1) introducing strong, but reversible albumin binding to reduce renal clearance and access to the insulin receptor and (2) drastically reducing the insulin receptor affinity as insulin is primarily cleared by internalization through its receptor. We hypothesised that ultra-long acting (referred to as ultra-long hereafter) insulin analogues would retain in vivo potency in spite of the low in vitro potency, albeit the in vivo effect would be executed over a prolonged time period.

**Structure of ultra-long insulin analogues.** The crystal structures of these low-affinity insulin analogues were determined to assess the effect of the substitutions and addition of albumin binders. The crystal structures of OI338 and OI320 shows a conserved insulin structure with small differences compared to the human insulin backbone, with root mean square deviations of 0.37 Å (OI338, 44C(alpha), Cα atoms) and 0.56 Å (OI320, 41 Cα atoms) (see Supplementary Fig. 2 and Supplementary Table 1).

**Stability towards proteolytic degradation.** OI106 (A14 Tyr→ Glu and B25 Phe→His) has 10-fold improvement in stability towards enzymes from rat duodenum relative to human insulin as shown in Fig. 2b. Deletion of B27 Thr (OI189) leads to additional 3–5-fold improvement.

Only limited differences in proteolytic stability are observed when comparing human insulin with OI216, OI106 with OI338 or OI189 with OI320 (Fig. 2b). Thus, the addition of the fatty acid moiety does not influence insulin stability towards degradation by these proteases.

**Insulin receptor affinity and in vitro potency.** The combination of amino acid substitutions and attachment of a strong albumin binder (C18 fatty diacid, linked via hydrophilic linker) to B29Lys reduces the insulin receptor affinity 500-fold compared to human insulin (in the presence of 1.5% HSA) (see Fig. 2a and c for structure and insulin receptor affinity). Furthermore, these insulin molecules with considerably reduced affinities for insulin receptors show proportionally low insulin-like growth factor 1 (IGF1) receptor affinities as compared to human insulin. Consistently, insulin receptor activation (tyrosine phosphorylation,

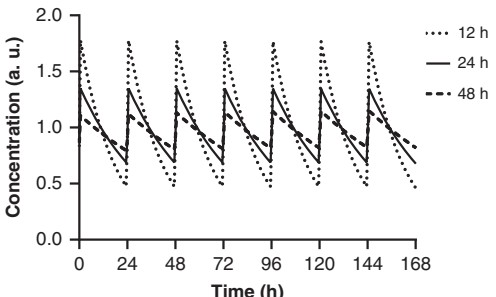

**Fig. 1 Influence of half-life on peak to trough in steady state in dogs with once daily dosing.** The simulations were based on data for OI338 tested in dogs and the PK profiles were normalized to the area-under-the-curve (AUC) for a 24-h period. Absorption and volume of distribution was kept constant and clearance was changed dependent of the elimination half-life.

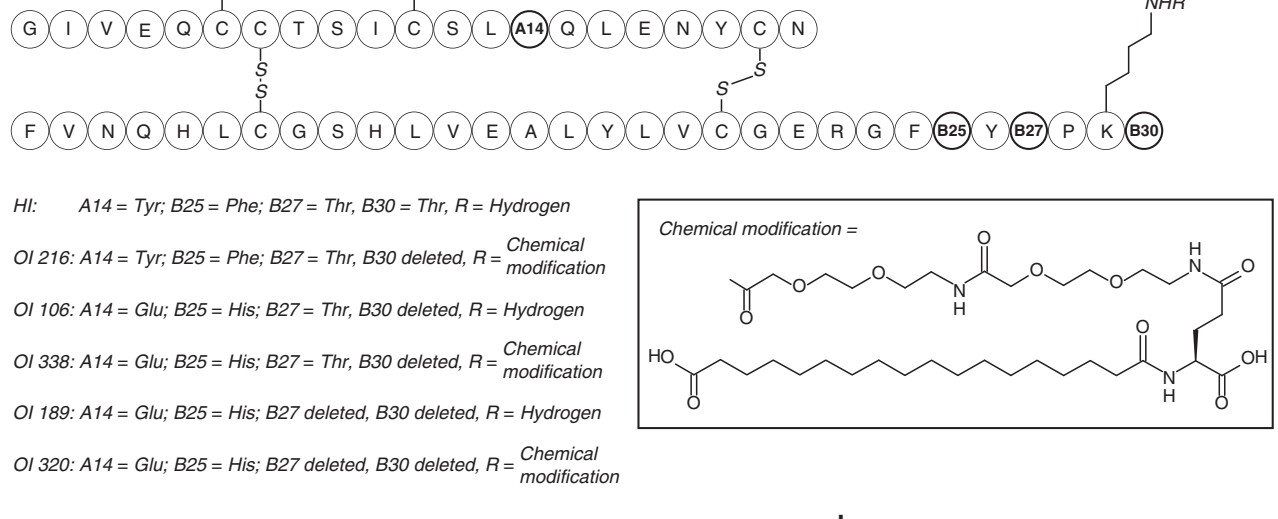

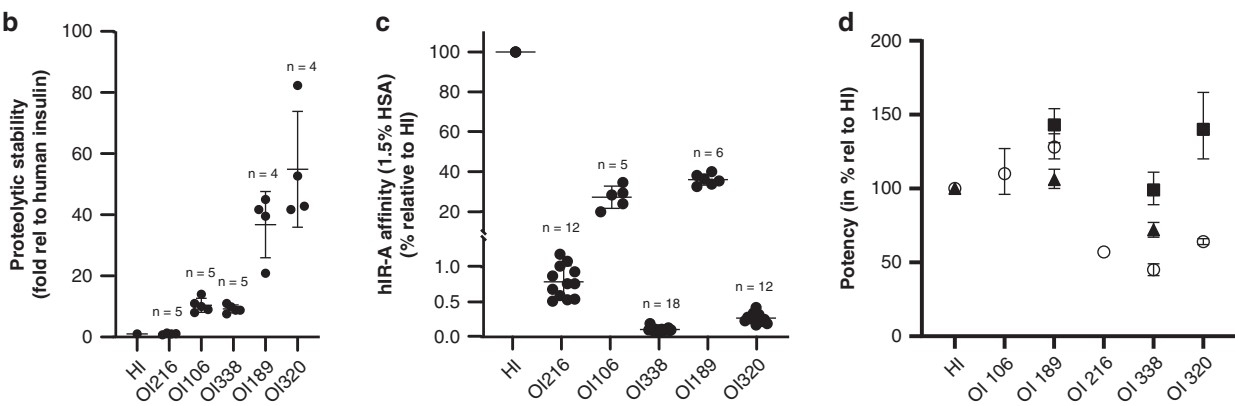

**Fig. 2 Structure and main biological properties of insulin analogues. a** Structures of the insulin analogues; note that R in HI, OI106, OI189 is hydrogen (i.e., chemically unmodified insulin) and that R in OI216, OI338 and OI320 is the chemical modification of B29 (see full drawing of chemical modification); Note that R is identical for OI216, OI338 and OI320 as illustrated in 2a) (all analogues are desB30). Insulin residues that are modified are A14 (Native Tyr, modified Glu), B25 (Native Phe), modified His), B27 Thr, modified by deletion (des), B30 Thr), modified by deletion (des). **b** Stability of the insulin analogues towards proteolytic degradation in rat duodenum extract. The degradation half-life relative to human insulin is shown (mean +/− SD from a number of independent experiments as indicated. **c** Receptor binding affinities of the insulin analogues relative to human insulin (HI = 100%) in the presence of 1.5% human serum albumin (HSA). Data are presented as mean values +/− SD from a number of independent experiments as indicated. **d** Potency of oral insulin analogues relative to human insulin on whole-body glucose disposal during different constant molar intravenous (IV) infusion rates to steady state (see the "Methods" section) under euglycemic clamp in healthy and conscious rats (circles), dogs (triangles) and pigs (squares), (confidence intervals is indicated and summaried in Table 1).

pTyr1158 hIR-A in CHO cells) and functional biological responses in primary adipocytes (lipogenesis) and hepatocytes (glycogen accumulation) from healthy rats are induced similar to human insulin, albeit with lower potencies reflecting the reduced insulin receptor binding affinities (see Supplementary Figs. 3 and 4, Supplementary Tables 2 and 3).

**GIPET®I tablets for oral dosing.** Despite substantial research interest in oral administration of protein and peptide drugs, few oral formulations have been developed for large animal models with translation possibility for clinical use. In this study, we used the GIPET®I tablet system, developed by Merrion Limited[11], based on sodium caprate as absorption enhancer. The GIPET®I tablet described in this study are identical to those used in the phase 2a clinical study[9] with regards to composition and size. The observed absorption profiles and bioavailability in both dogs and human are comparable albeit the insulin dose was lower in the dog studies. Improvement of the GIPET®I formulation was not the aim of this study.

**Pharmacokinetics and bioavailability in Beagle dogs.** The reduced insulin receptor affinity increases the intravenous (IV) half-life in dogs from 25 min (human insulin), to 24 h (OI338) and 17 h (OI320), respectively. The strong binding of these two representative insulin analogues, OI338 and OI320, to albumin (Supplementary Fig. 5) ensures that the majority of the molecules in circulation are bound to albumin.

As shown in Fig. 3a, stabilisation of the insulin backbone as well as the attachment of the fatty acid-based side chain contributed to effective plasma exposure after oral dosing. Characterization of pharmacokinetic profiles (after oral dosing in GIPET®I) shows an absolute bioavailability (F) of 3.0% for OI338 (CV 100%, single dose, $n = 40$) and 3.9% (CV 78%, single dose, $n = 63$) for OI320 (Fig. 3c), calculated relative to IV dosing. OI216 (human insulin desB30 back-bone) with an identical fatty acid side chain to OI338 and OI320 has much lower bioavailability, emphasising the importance of stabilisation of the insulin backbone (Fig. 3c). Furthermore, the oral bioavailability of insulin aspart, a marketed rapid-acting insulin analogue without

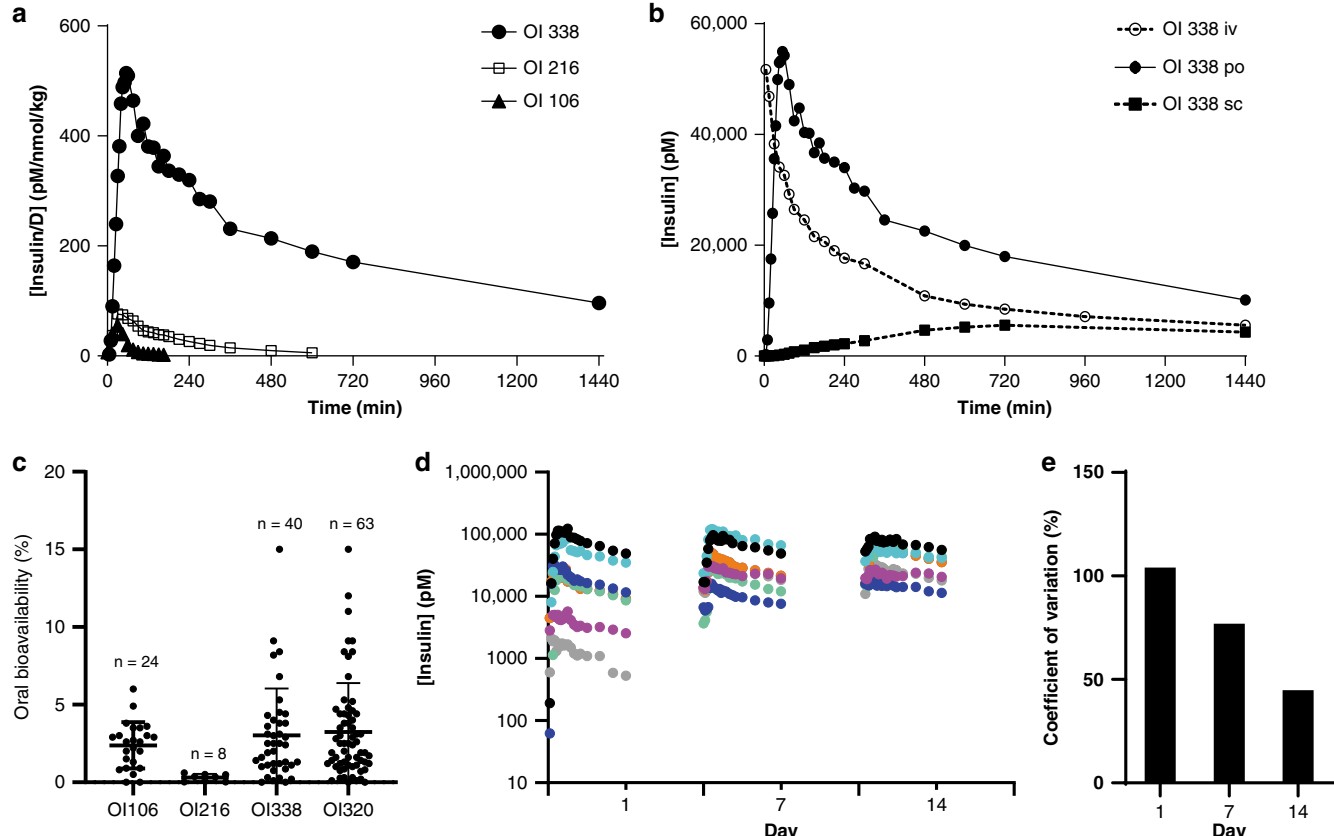

**Fig. 3 Pharmacokinetic properties of oral insulin analogues. a** Pharmacokinetic profiles of OI106 (triangles), OI216 (squares) and OI338 (circles) after single oral dosing in healthy and conscious dogs ($n = 8$); the plotted concentrations are exposure divided by the oral dose in nmol/kg for each insulin analogue. Coefficient of variation, CV, for the AUC, was 75–100%. **b** Pharmacokinetic profiles of in OI338 after IV (2 nmol/kg, $n = 8$); peroral (105 nmol/kg, $n = 8$) and SC (4 nmol/kg, $n = 4$) administrations in dogs. CV for AUC was around 10% after IV and SC administration and ~100% after oral dosing. Note the fast absorption profile after oral administration (closed circles). After reaching $C_{max}$ the oral pharmacokinetic profile almost mimics that of an IV administration (see also Table 1). **c** Bioavailability of different insulin analogues after oral administration in GIPET®I tablets to dogs (mean +/− SD from a number of independent experiments. **d** Individual 12-h pharmacokinetic profiles (day 1, 7, 14) of OI338 after multiple oral daily dosing in eight dogs (each dog one colour). **e** Coefficient of variation of bioavailability based on area under the curves in Fig. 3d), multiple dose study. Intra/inter-individual CV based on the mean of the group was reduced from 100 to 40% after treatment to steady state.

stabilising substitutions or fatty acid moiety was also noticeably lower, 0.63% (CV 143%, $n = 24$).

The pharmacokinetic profile for OI338 after oral dosing was characterised by a rapid absorption period, leading to a steep increase in plasma concentration and reaching an early $T_{max}$ which resembles the IV pharmacokinetic profile (Fig. 3b). SC administration, on the other hand, leads to a slow absorption with a much later $T_{max}$ as well as lower exposure level (Fig. 3b). A multiple dose study wherein OI338 was dosed daily orally to dogs for 14 days demonstrates that accumulation to a steady-state exposure occurs after 5 days. The intra-/inter-individual coefficient of variation, determined as the within day variation observed in the group, was found to be reduced over time as can be seen by less peaked individual PK profiles and less variation between dogs on day 14 (40%) compared to day 1 (100%) (Fig. 3d, e). In man, in an 8-week study, the variability observed for fasting plasma glucose for the OI388 group is low with a coefficient of variation of 23.2% after oral administration which is similar to the 17.5% for insulin glargine after SC administration[9]. General pharmacokinetic parameters in healthy and conscious Beagle dogs are summarised in Table 1).

**Pharmacodynamic potency as whole-body glucose disposal.** Importantly, these low-affinity insulin analogues retain whole-body glucose disposal potencies comparable to human insulin as

determined in healthy and conscious dogs, pigs and rats under hyperinsulinemic euglycemic clamp conditions, despite their apparent low in vitro potencies (Fig. 2d). A representative dose/response set from a rat clamp study is shown in Fig. 4. The sigmoidal log(dose) response curves for OI320/OI338 are right shifted compared to that of human insulin but the maximal efficacy on glucose infusion rate (GIR) is similar for healthy and conscious rats, pigs and dogs. At least three different doses (constant molar infusion rates) of human insulin and the oral insulin analogue in question were tested in the dose response glucose clamp studies in the three species. The doses were chosen individually in the different species to be at the linear part of the log(dose) response curve based on historical data for human insulin and on a priori assumptions for the oral insulin analogues. See Table 1 for a summary of relative in vivo potencies on whole-body glucose disposal.

**Steady-state distribution of ultra-long insulin analogues.** When administered SC, insulin is absorbed into the peripheral circulation and distributed in equal arterial concentrations to all target tissues. This contrasts with endogenously secreted insulin which reaches the liver as the first target organ in higher portal concentrations and is subject to first-pass clearance before it reaches the peripheral target tissues, such as muscle and fat, in lower concentrations. This has been demonstrated in healthy conscious

**Table 1 Pharmacokinetic and pharmacodynamic parameters of insulin analogues in animals.**

| | Pharmacokinetic parameters in the dog[a] | | | Potency on whole-body glucose disposal relative to human insulin[b] (%) | | |
|---|---|---|---|---|---|---|
| | $T_{max}$ min | $T\frac{1}{2}$ dog oral (h) | $T\frac{1}{2}$ dog iv (h) | Rat | Pig | Dog |
| Human insulin | | | 0.25 ± 0.1 | | | |
| OI216 | 30 | 3.8 ± 1.9 | 5.7 ± 0.3 | 57 [54–60] | | |
| OI338 | 45 | 17 ± 2 | 24 ± 0.4 | 45 [43–47] | 99 [89–110] | 72 [58–86] |
| OI 106 | 45 | 0.7 ± 0.3 | 0.8 ± 0.02 | 110 [105–115] | | |
| OI 189 | | | | 128 [123–134] | 135 [94–203] | 106 [100–117] |
| OI320 | 50 | 14 ± 2 | 17 ± 1 | 64 [62–66] | 140 [120–170] | |

[a]The pharmacokinetic parameters are from non-compartmental analysis of individual profiles from either IV or oral bolus studies in dog. $n = 2$–4 for the IV studies and $n = 8$ for the oral studies. Mean ± SD is given for half-life and median for $T_{max}$.
[b]Potency relative to human insulin at steady state during hyperinsulinaemic euglycaemic clamp (95% confidence interval).

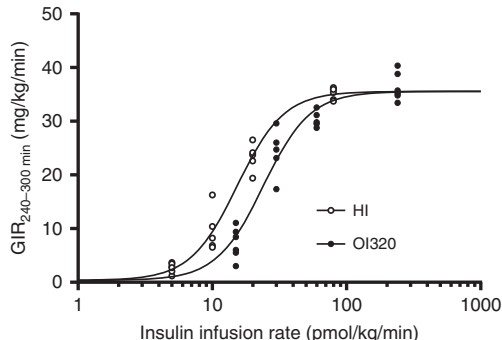

**Fig. 4 Hyperinsulinemic, euglycemic clamp experiment.** A representative dose response plot (GIR = glucose infusion rate) from a 5-h hyperinsulinemic, euglycemic clamp experiment in healthy and conscious rats performed to determine the potency of OI320 (black circles) relative to human insulin (white circles). The dots represent the average GIR during the last 60 min of the experiment in each rat ($n = 6$ rats per dose (group); independent experiments). Similar dose response studies were performed in healthy and conscious pigs and dogs as indicated in (Fig. 2d) and; see Table 1 for 95% confidence interval.

dogs by comparing intraportal versus peripheral venous infusions of human insulin, resulting in insulin concentrations in the peripheral circulation that are ≈20% higher than in the portal vein after peripheral venous infusions[15]. The ultra-long insulin analogues are not expected to have noticeable first-pass clearance due to their low receptor affinity.

We modelled the steady-state distribution between the central (systemic circulation/plasma, including the liver) and peripheral (tissue/interstitial fluid, including metabolically relevant tissues for insulin such as fat and muscle) compartments to assess differences between orally administered OI338 and SC delivered human insulin (Supplementary Fig. 6). The simulations showed that a balance between elimination half-life, slow peripheral distribution and the oral administration route leads to an initial peak OI338 in the central compartment and a delayed appearance in the peripheral compartment due to slower trans-endothelial transport of the albumin bound insulin analogues. In contrast, simulations of human insulin dosed SC show a more pronounced peak in the peripheral compartment (Supplementary Fig. 6). Thus, the simulations predict that the rapid absorption of OI338 into the portal vein when dosed orally will lead to a different tissue distribution compared to human insulin dosed SC, i.e., more restricted to the central compartment in the initial phase.

In addition to the advantages that this type of distribution would have to minimize the risk of hypoglycaemia due to

variations in oral absorption (see section below on hypoglycaemia assessment), it could be speculated that an insulin analogue with strong albumin binding having a rapid, peaked central (blood compartment) PK profile would mimic the pattern of insulin secreted from the pancreas directly into the portal circulation where the liver is highly exposed. This peaked PK profile upon oral delivery with greater liver effect is not expected to be the same as what was observed for insulin PEGlispro which demonstrated a constant hepato-selective effect[16]. Whether this more physiological profile of rapid and peaked insulin delivery to the liver (part of the blood compartment due to the fenestrated capillaries) would lead to improved metabolic control remains to be determined. Similarly, this type of insulin analogue may be expected to have an altered CNS distribution and it is not known whether this would lead to any differences in central effects compared to human insulin as has been suggested for insulin detemir[17].

**Pharmacokinetic and pharmacodynamic relationship.** To test the accuracy of the simulations regarding the distribution of OI338 and OI320, the link between dynamic changes in the pharmacokinetic profile and the pharmacodynamic response were studied in healthy and conscious dogs during euglycemic clamp settings after a single oral administration of OI320 (Fig. 5d). OI320 showes a prolonged lag time of 6–8 h between the plasma $C_{max}$ and maximal pharmacodynamic effect as quantified by the GIR required to maintain euglycemia. The lag time between $C_{max}$ and maximal GIR may be attributed to the delayed appearance of OI320 in the peripheral compartment where muscle and fat tissue are responsible for insulin-stimulated glucose uptake.

**Assessment of hypoglycaemia after overdosing.** Next, we addressed the relevant safety concerns related to the day to day variability in orally administered insulin absorption. Healthy and conscious mongrel dogs were pre-treated once daily with OI320 for three days to reach steady state and on the fourth day, OI320 was administered via the portal vein by a 40 min square wave infusion to mimic an oral absorption with up to three times the daily maintenance dose to study the hypoglycaemic response and dynamics of glucose turnover. The baseline OI320 concentration on the fourth day, 24 h after the third maintenance dose was 8009 ± 599 pM (mean ± SEM for all animals). OI320 lowered plasma glucose dose dependently; the high dose levelled around 3 mM whereas the two lower doses plateaued around 6 and 4 mM (Fig. 6a). As plasma glucose decreases, initially due to inhibition of glucose production (Ra) by the liver, glucose disappearance (Rd) also goes down slightly, indicating that there is no

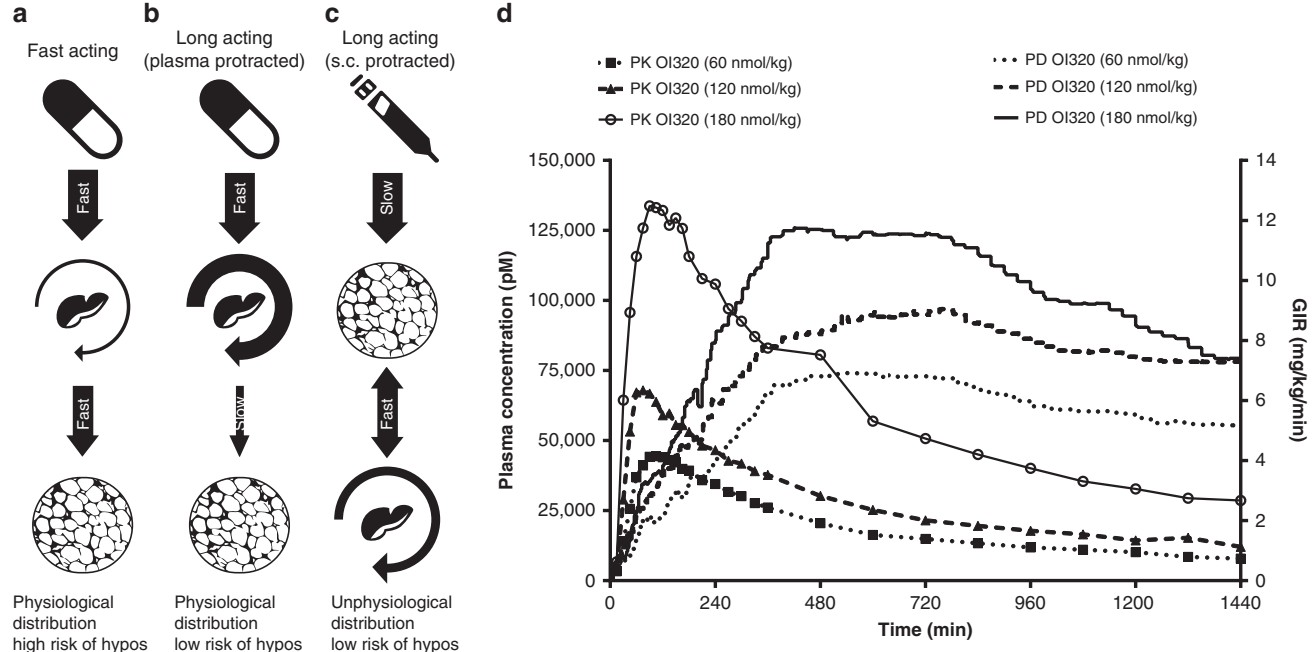

**Fig. 5 Schematic overview of insulin levels, distribution kinetics and the kinetic/dynamic relationship. a–c** Schematic overview of insulin levels (arrow width) and distribution kinetics upon **a** oral dosing of fast acting insulin (e.g. human insulin), **b** oral dosing of a plasma protracted insulin (e.g. OI338 and OI320), and **c** SC dosing of a long acting insulin, which is protracted SC (e.g. insulin glargine or insulin degludec, or OI338/OI320 dosed SC). **d** The pharmacodynamic time course (GIR = glucose infusion rate required to maintain euglycemia) of orally dosed OI320 and pharmacokinetic exposure during hyperinsulinemic euglycemic clamp in healthy and conscious dogs. Note the long lag time of pharmacodynamic effect in respect to exposure. The CV for the GIR AUC was in the range 45–80% and for the pharmacokinetic AUC the CV was 45–100%.

stimulation of Rd by OI320 (evident by unchanged glucose clearance during the first 30–45 min, Fig. 6d). However, after 60 min Rd increases again and plateaus close to baseline. Since plasma glucose concentrations at the high OI320 doses are only half of baseline, i.e., glucose clearance has doubled, this indicates a peripheral insulin effect after one hour. In other words, Rd was affected by both glucose and insulin—the lower glucose levels bringing it down; the higher OI320 levels bringing it up.

In contrast, over-dosing with human insulin (60% of the OI320 maintenance dose) quickly affects both glucose Ra and Rd, leading to acute hypoglycaemia and requiring intervention with glucose infusion to prevent plasma glucose levels from dropping below 2.5 mM (Fig. 6a). Ra actually increased which is likely a response to the decrease in plasma glucose caused by the rapid stimulation of Rd (Fig. 6b, d). The lag time between $C_{max}$ of human insulin when infused into the portal vein for 45 min in dogs and time for maximal glucose lowering (Fig. 6a) is 0–15 min (plasma exposure shown in Supplementary Fig. 7b) thus shorter than the long lag time observed for OI320 (Fig. 5d). Thus, the delayed distribution of OI320 to the peripheral tissues after portal dosing blunts hypoglycaemia in response to overdosing in healthy and conscious dogs (additional data on arterial insulin concentrations, C-peptide, glucagon and epinephrine are available in Supplementary Fig. 7). This comparison to human insulin highlights the advantage of oral delivery of insulin analogues such as OI320 or OI338 with a more restricted distribution within the central/blood compartment in order to reduce the risk of hypoglycaemia that may occur with variations in oral absorption and at the same time would caution against the use of regular human insulin or rapid-acting insulin analogues for oral therapy. In people with type 2 diabetes, orally administered OI338 had an elimination plasma half-life of 70 h at pharmacokinetic steady state and a relative bio-efficacy of 1.7% compared to SC administered insulin glargine, with no additional risk of

hypoglycaemia[9]. Recent technological developments within the field of oral delivery of macromolecules, and particularly progress within oral delivery devices[18] together with the tailor-made insulin molecules outlined here, may enable oral basal insulin therapy in the future.

## Methods

**Production of insulin analogues.** Insulin precursors were expressed in *Saccharomyces cerevisiae* (*MAT*a/*MAT*alpha, *leu2/leu2*, *TPI1::LEU2/TPI1::LEU2*, *HIS4/his4*, *pep4-3/pep4-3*, *Cir+*) as proinsulin-like fusion proteins consisting of the pre-pro-segment of the yeast mating pheromone alpha-factor, a spacer (EEAEPK) followed by the B-chain (B1-B29) linked to the A-chain (A1-A21) by a mini C-peptide AAK[19]. The single-chain precursors were enzymatically converted into mature two-chain desB30 insulin analogues using *Achromobacter lyticus* protease I (EC 3.4.21.50) specific for lysyl peptide bonds. The insulin analogues were chemically modified by coupling of [2-(2-[2-(2-[2-(octadecandioyl-γGlu)amino] ethoxy)ethoxy]acetylamino) ethoxy]ethoxy)acetyl-OSu] to the epsilon amino group of B29 lysine[20,21]. In brief, the N-hydroxysuccinimide activated side chain was dissolved in DMF and added slowly to a well stirred solution of insulin (100 mg/mL) in 0.1 M $Na_2CO_3$ at 0–5 °C, while keeping pH at 11 by simultaneous addition of 1 M NaOH. The product was purified by Reverse-Phase High-performance liquid chromatography (RP-HPLC) and isolated by lyophilization. All insulin analogues are lacking B30 threonine except insulin aspart.

**Proteolytic degradation.** Degradation of insulin analogues was performed using duodenum lumen enzymes (prepared by filtration of duodenum lumen content) from healthy Sprague Dawley (SPD) rats[22]. Insulin analogues ~15 μM was incubated with duodenum enzymes in 100 mM Hepes, pH = 7.4 at 37 °C, samples were taken after 1, 15, 30, 60, 120 and 240 min and reaction quenched by addition of trifluoroacetic acid. Intact insulin analogues at each point are determined by RP-HPLC. The amount of enzyme added for the degradation was such that the half time for degradation cof the reference insulin was between 5 and 15 min. The result is given as the degradation half time for the insulin analogue in rat duodenum divided by the degradation half time of the reference insulin from the same experiment (relative degradation rate).

**Tablet formulation.** Tablets were prepared according to standard pharmaceutical techniques in Good Laboratory Practice production laboratories. The tablet core composition was as follows for a final weight of 710 mg/tablet: Active ingredient

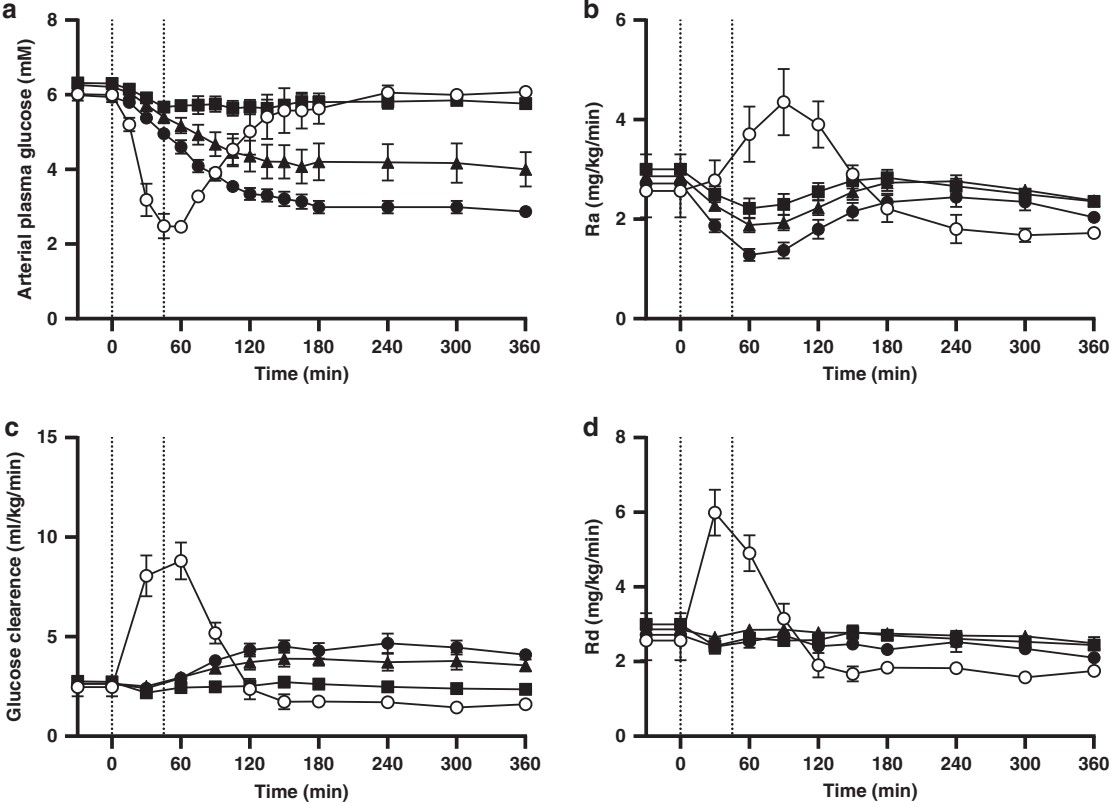

**Fig. 6 Assessment of hypoglycaemia after overdosing in dogs.** Plasma glucose lowering, and dynamics of glucose turnover obtained from an over-dose study with OI320 performed in healthy and conscious mongrel dogs on day four after once daily IV dosing with a maintenance dose of OI320 (40 pmol/kg/min for 45 min) for three days. The three days of pre-treatment was performed to mimic a steady-state situation. **a** The hypoglycaemic response in dogs to 45 min intraportal infusion of IO320 in three different doses (40 (black squares), 80 (black triangles) or 120 (black circles) pmol/kg/min) ($n = 5$ for each doses; independent experiments; mean +/− S.E.M.) or human insulin (white circles) at 24 pmol/kg/min ($n = 3$; independent experiments; mean +/− S.E.M.; independent experiments). The dose of human insulin was set to 60% of the maintenance dose of OI320 (40 pmol/kg/min) in order not to induce too severe hypoglycaemia. **b** The dynamics of glucose appearance (Ra, hepatic glucose production) and **d** glucose disappearance (Rd, glucose uptake mainly by muscle and fat). **c** Changes in glucose clearance. The vertical lines in Fig. 6a–d indicate the portal insulin infusion period from 0–45 min.

(dose dependant), Sodium caprate 550 mg, sorbitol (149 mg minus insulin dose) and 3.5 mg stearic acid. The tablets were compressed on a Fette 102i tablet press from Fette Compacting equipped with a 16-punch turret. For the tablet cores a set of oblong tablet punches (double radii convex 9 mm × 18 mm) were used. All tablets, except of OI106, were finally film-coated with 4.5% (w/w) Opadry (Polyvinyl alcohol) in an O'Hara Labcoat drum coater. All tablets tested met all compendial rules for content uniformity and technical properties according to the United States Pharmacopoeia as well as the European Pharmacopoeia. The free acid of sodium caprate—capric acid constitute close to 10% in coconut oil and up to 1% in milk and it's generally recognized as safe. As a food additive it is regulated in the US under the Electronic Code of Federal Regulations, e-CFR chapter 172.860.

**Three-dimensional structural determination.** Insulin crystals were grown using the sitting drop vapour diffusion technique where equal volumes of insulin and precipitant were incubated over precipitant. OI338 (25 mg/ml in water, pH 7.5, added 0.1% (w/v) NVoy (Expedeon)) was crystallized using 6.0 M ammonium nitrate, 0.1 M Tris pH 8.5 as precipitant (0.5 + 0.5 µl drops incubated over 100 µl precipitant at 37 °C). OI320 (8 mg/ml in water, pH 7.5) was crystallized using 0.12 M monosaccharides, 0.1 M Buffer System 1 pH 6.5, 20%(v/v) PEG500MME, 10%(w/v) PEG20,000 (F1 from Morpheus screen, Molecular Dimensions[23]) as precipitant (0.36 + 0.36 µl drops incubated over 80 µl precipitant at 20 °C).

**Diffraction data collection, structure determination and refinement.** Diffraction data to 1.5 Å resolution were collected at 100 K on crystals first cryo-cooled in liquid nitrogen and the OI338 crystal was beforehand cryo-protected in precipitant with 20% ethylene glycol added. The OI338 data were obtained on a CCD165 detector (Mar Research) mounted on a MicroMax-007 HF rotating anode (Rigaku) using CuKa radiation (1.5418 Å). The OI320 data were collected on a Pilatus 2 M detector (Dectris) at beamline X06DA, Swiss Light Source using a wavelength of 1.000 Å. The diffraction data were auto indexed, integrated and scaled using

programs from the HKL2000[24] (OI338) and XDS[25] (OI320) suites. The OI338 crystal structure was phased by single wavelength anomalous dispersion using the sulfur atoms from insulin's three disulfide bridges in the Autosol module of the program suite Phenix[26] where automated model-building also was applied. The OI320 crystal structure was determined with molecular replacement in the program Phaser[27] implemented in Phenix using the OI338 crystal structure as search model. Both crystal structures were refined in Phenix with rounds of manual model building in Coot[28]. The Ramachandran plots of both structures have 98% of the amino acid residues in the favoured areas and the remaining 2% in the allowed areas. The final diffraction data and refinement statistics are summarized in Supplementary Table 1.

**Pharmacokinetic studies in dogs**. All animal study protocols, except those involving mongrel dogs, were approval by the Danish Animal Experiments Inspectorate. The mongrel dog protocol was approved by the Vanderbilt University Institutional Animal Care and Use Committee, and housing and care followed American Association for Laboratory Animal Care guidelines. Adult male mongrel dogs (21–24 kg) purchased from a U.S. Department of Agriculture–licensed vendor was studied. All animal study protocols were approved by Novo Nordisk Ethical Review Committee, Denmark. All animal protocols are in compliance with all relevant ethical regulations. Healthy and conscious Beagle dogs (males ~ 17 kg) were fasted overnight with access to drinking water. In the morning the dog was placed on a test platform and a Venflon 20 G was inserted in the *v. cephalica* for blood sampling during the first 6 h post-dosing. Thereafter, the venflon was removed and the dogs were returned to their booths and were offered food and exercise in the outside run. At the later time points the dogs were led into a test room for blood sampling directly from the jugular vein using a needle and syringe.

Peroral (PO) administration: Dogs were dosed PO with the OI338/OI320/OI216/insulin aspart GIPET®I tablets. After administration of the tablet the dog was given 10 ml drinking water directly into the mouth by a syringe to make sure that the tablet had entered the oesophagus. Dogs were offered small walks when needed for urination and offered drinking water 120 min after the tablet was dosed.

IV administration: Insulin was administered as an intravenous bolus via a Venflon 20 G inserted in the *v. cephalica* of the leg opposite to that of the sampling Venflon.

SC administration: The day before the experiment the fur on a predefined area (~2 cm in diameter) of the shoulder was removed and the test compound was administered SC using a NovoPen®Echo.

**Glucose clamp studies in Beagle dogs**. Healthy and conscious Beagle dogs (males ~ 17 kg) were fasted overnight with access to drinking water. In the morning the dog was placed on a test platform and shortly sedated while a temporary triple lumen catheter for blood sampling and infusion was inserted in the jugular vein after which the sedation was reversed. Plasma glucose concentrations were measured at various intervals as required to maintain a euglycaemia clamp using an auto-analyser (YSI 2300D Stat Plus, Yellow Springs Instruments Co., Inc., Ohio, USA), and the GIR was adjusted accordingly to maintain euglycemia. Insulin was either infused IV by primed, constant rates for determination of potency relative to human insulin (three dose levels) (Fig. 2d and Table 1) or administered PO in GIPET®I tablets (Fig. 5d).

**Studies in mongrel dogs**. Dogs were housed according to Association for Assessment and Accreditation of Laboratory Animal Care International guidelines in a USDA-inspected facility on a 12 h/12 h light/dark cycle with male companions. Once daily, dogs were fed a weight-maintaining diet providing 28% protein, 49% carbohydrate, and 23% fat, consisting of Lab Diet canine 5006 and Pedigree Choice beef cuts in gravy[29]. Licensed veterinary staff from the Vanderbilt University Medical Center supervised their care and evaluated health. Dogs were fasted for 18 h before each study. For oral insulin administration studies, eight healthy and conscious dogs were trained to take an oral placebo tablet for 2 weeks before study. On the morning of study, an active insulin tablet (OI338 or OI320) was administered orally at a dose of either 120 or 180 nmol/kg. Venous insulin concentrations were monitored for 72 h after dosing, with normal food administration resuming 6 h after each insulin dose. All dogs underwent four studies at 1-week intervals, receiving both analogues at both dosages with the order of analogue administration/dosage randomized within the group.

For dogs undergoing pancreatic clamps, surgery was performed under general anaesthesia approximately 16 days before the study to insert sampling catheters in the femoral artery, hepatic portal vein, and left common hepatic vein; blood flow probes around the portal vein and hepatic artery; and splenic and jejunal vein catheters to allow infusion into the hepatic portal circulation[29]. On the morning of study, the proximal ends of the catheters and flow probes were removed from their SC pockets under local anaesthesia and dogs were placed in a Pavlov harness. 3H glucose was infused via peripheral vein[29]; after basal sampling (−20 to 0 min); insulin was infused into the portal circulation via the splenic and jejunal catheters from 0–45 min, and sampling continued until 360 min. Dogs received analogue OI320 at a rate of 40, 80, or 120 pmol/kg/min ($n = 5$/dose) or human insulin at 24 pmol/kg/min ($n = 3$). Glucose was delivered via peripheral vein as necessary to maintain arterial glucose ≥ 2.5 mmol/L.

**Euglycaemic clamp in pigs**. Healthy and conscious female domestic pigs (Landrace × Yorkshire × Duroc (LYD)) were used for the glucose clamp studies employing a cross-over design. In brief, human insulin or primed oral insulin analogues was infused IV at three different constant molar rates and blood samples were drawn frequently for measurements of plasma glucose concentrations (YSI 2300 Stat Plus, Yellow Springs Instruments Co., Inc., Ohio, USA) while a glucose solution was infused at variable rates (0–24 h) to maintain plasma glucose at the clamp level[30].

**Euglycaemic clamp in rats**. Healthy male Sprague Dawley rats were prepared for Euglycaemic clamp[31]. In brief, human insulin or primed oral analogues were infused IV at three to four different constant molar rates and blood samples were drawn frequently for measurements of plasma glucose concentrations (YSI 2300 Stat Plus, Yellow Springs Instruments Co., Inc., Ohio, USA) while a glucose solution was infused at variable rates (0–5 h) to maintain plasma glucose at the clamp level.

**Calculation of in vivo potency and confidence intervals**. The log transformed insulin infusion rates (pmol/kg/min) and mean of the corresponding GIR (mg/kg/min) during the last 60 min of the euglycaemic clamp in each animal were fitted to Sigmoidal dose-response curves and the potency of the oral basal insulin analogues relative to human insulin was calculated as the ratio between their ED50% values using GraphPad Prism's (version 8.0.2 for Windows) log(Agonist) vs. response— Variable slope (four parameters) fitting model (Eq. (1)):

$$Y = \text{Bottom} + (\text{Top} - \text{Bottom})/(1 + 10 \wedge ((\text{LogED50} - X) * \text{HillSlope})) \quad (1)$$

Fitting constrains were shared values for Hillslope, maximal GIR response (Top) and the GIR response to no insulin infusion (Bottom). The 95% confidence interval of the ratio between means of the ED50% was calculated using Fieller's Theorem[32].

**Quantification of insulin concentration in plasma samples**. For pharmacokinetic assessment blood samples were drawn and insulin plasma concentrations were analysed using luminescence oxygen channelling immunoassay[33,34] using suitable monoclonal antibodies. The lower limit of quantification of, e.g., OI320 in dog plasma was 50pM. Regarding specificity, the cross reactivity of antibodies to canine insulin was <1%.

**Reporting summary**. Further information on research design is available in the Nature Research Reporting Summary linked to this article.

## Data availability
Crystal structure coordinates and structure factors were deposited in the Protein Data Bank (PDB) under the following accession codes: 6S4I (OI338) and 6S4J (OI320). The authors declare that the data supporting the findings of this study are available within the paper and its Supplementary information files. Data are available from the corresponding author upon request.

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

## Acknowledgements
The authors wish to thank Annette F. Bjerre, Lene Walander, Lone Honoré, Stine V. Johansen, Dorthe E. Jensen, Gitte Norup, Anne-Lise R. Gudmundsson, Carsten S. Stenvang, Jane K. Gravsen, Anja Benfeldt, Sandra B. Jensen, Nete Ibsen, Thorbjørn N. Richter-Olesen, Susanne Gronemann, Brian S. Hansen, Jeanette Hansen, Iben T. Nielsen, Bettina L. Bengtsen, Merete Hvidt, Birgitte Roed, Tina K. Rasmussen, Pia Jensen, Irena Duch, Marianne Schiødt, Mette Frost, Lenette S. Jørgensen, Kirsten Vestergaard, Birgit Klaproth, Gitte-Mai Nelander for technical assistance. Furthermore, we acknowledge Merrion Limited for development of the GIPET®I oral delivery platform. Valuable comments to the paper by Dr. Jeppe Sturis are appreciated. We are grateful to Janos Tibor Kodra and Jette Lenstrup, both of Novo Nordisk A/S, who conceived and provided the OI216 insulin analogue. Furthermore, we are appreciative of Inger Lautrup-Larsen, Novo Nordisk A/S, contributions with recombinant expressing, as well as many colleagues' producing, purifying and quantifying the compounds used in this study.

## Author contributions
F.H., P.K.N., S.L., P.M., P.G.B. and T.K. designed insulin analogues. F.H. and P.K.N. established and executed proteolytic stability screening. D.B.S. and S.L. designed and performed solubility screening. E.J. prepared crystals and solved 3-D structures. T.G. and T.K. established and executed receptor affinity screening. C.E.S. measured albumin binding. C.E.S., E.N., B.F.H., and M.K.J. characterized insulin analogues in cellular assays. L.H. designed, optimized and prepared insulin GIPET®I tablets. T.P. and C.L.B. designed and executed rat experiments. S.G.N. and E.M.W. designed and executed beagle dog studies. J.K. and U.R. designed and executed LYD pig studies. M.C.M. carried out the canine studies at Vanderbilt University (Fig. 4 and a portion of Table 1). M.C.M and A.D.C. participated in the design of those experiments and analysed and interpreted data. H.H.F.R., L.P. and C.H.C. designed and executed PK calculations, modelling and simulation. F.H., H.H.F.R., L.P., P.M., S.L., S.R. and T.K. led and coordinated the studies. F.H., H.H.F.R., M.M., C.L.B., E.N. and T.K. drafted the manuscripts. All authors read and contributed to editing of the paper.

## Competing interests
All authors except M.C.M. and A.D.C. are present or past employees of Novo Nordisk and hold equity in the company. M.C.M. and A.D.C. received financial support from Novo Nordisk. Insulin analogues and formulations described in this article are covered by Novo Nordisk's patents and patent applications. A.D.C. reports grants, personal fees, and other from Metavention, Zafgen, and Abvance; grants and/or personal fees from Thetis Pharmaceuticals, Boston Scientific, Novo Nordisk, vTv Therapeutics, Merck, Eli Lilly, and Galvani Bioelectronics; personal fees from California Institute for Biomedical Research (Calibr) and MedImmune; personal fees and other from Fractyl, Biocon and Sensulin Labs.
