## [Peer Review File · Nature Communications]

Reviewers' Comments:

Reviewer #1:

Remarks to the Author:

In the communication from Kjeldsen's group, the development of ultra-long acting insulin analogs for the use of oral insulin therapy for diabetic patients has been reported. This is achieved by the combination of the pertinent mutations on the amino acid sequence of human insulin with the modification at the B29 lysine residue with an albumin-binding fatty acid. These modifications indeed increase the life-time of the bioavailability in the plasma significantly. The achievements might have a strong impact in the communities. However, in consideration that one of the oral insulin analogs, i.e., OI338, was already reported and tested for the pharmacokinetic analysis for men in refs 1 and 2, where conceptually the same strategy was employed for the development of oral insulin drugs, scientific advances of the submitted paper would be rather limited. Although several new experimental data, including new ultra-long acting insulin analogs, such as OI216 and OI320, the crystal structures, and their detailed pharmacokinetic and pharmacodynamic behaviors for dogs, have been presented, the paper is not recommended for publication in Nature Communications due to the lack of enough novelty and innovative data.

Reviewer #2:

Remarks to the Author:

In this study, the authors revealed that the combination of changes in three different properties of human insulin considerably increased the feasibility of developing oral insulin. The changes include enhancing the proteolytic stability, increasing the albumin binding affinity and decreasing insulin-receptor binding affinity. In addition, they found that the new engineered molecules may provide direct protection against hypoglycemia, a condition that is usually resulted from insulin overdose and can cause severe health consequences. This research is original and important. The research design is well developed, and the experimental procedures and results are clearly described. The data from this research study support the feasibility of developing oral insulin for diabetes treatment. This work is expected to quickly catch the attention of scientists in the field of insulin. Overall, this manuscript is within the scope of Nature Communication and I recommend publication as it is.

Reviewer #3:

Remarks to the Author:

Nat Com 235956; Hubálek et al

The paper from Hubálek and colleagues et al is an interesting description of an approach to design of insulin analogues for oral administration. The topic of high interest, although whether or not these efforts lead to a marketable product is a subject for future clinical work. The manuscript is dense, but goes to some trouble to explain the rationale of the steps described, and does this with useful clarify.

My comments are generally editorial:

1. The authors vary in their usage of types of abbreviation for amino acids, in a way that is not good communication and can be confusing. At a simple level most clinical readers will not be familiar with the single letter format, and that is not intuitive. In Figure 2a it is confusing, as both notations are used, and 'H' is used for hydrogen and for histidine and for human! The reviewer suggests revising to the 3-letter form for amino acids throughout.
2. Figure 2 is not intuitively laid out in any case. It took the rather slow reviewer some 3 minutes to work out that the three 'R =' usages without anything following referred to the aliphatic side chain displaced further to the right and somewhat lower. Using a brace (curly bracket) with the chemical formula immediately adjacent to it would probably largely resolve the problem.
3. In any case what is the significance of the red coloured symbols in this chemical structure? This

is not explained in the legend.

4. Again while 'Ca atoms' will be known to the authors, the reader unfamiliar with the analogues and size estimates will wonder if calcium is really meant. Suggest spell it out on first usage. Oddly zinc is always spelt out in the supplementary material.

5. Sodium caprate will not be familiar to many. It is briefly introduced and referenced however. It is not a US FDA GRAS substance I believe. Perhaps it is already used in some other orally administered substances (apart from being present in some foods). Some statement on its regulatory status in Europe and the US would be welcome.

6. In the Abstract the authors comment on 'efficient' glucose lowering in animals in clamps. While it is obvious to the authors, some readers may not realize that IV administration is used in these studies. It is suggested to say that.

7. It is anyway not clear how the IV dosing relative the human insulin was arrived at. To this reviewer efficiency would imply that, at steady state with albumin binding saturated, the dose giving 100 % equivalent in figure 2 would have the same molar content as the human insulin standard. Was that the case? Otherwise efficacy is established not efficiency. Figure 4 on the horizontal axis is indeed molar (and shows a halving of potency, which is indeed a minor difference), but the authors could be more explicit in the relevant parts of the text as to doses used.

8. In Figure 3 the oral doses are described as 'dose normalized'. This might mean just about anything, but perhaps means doses were adjusted for the oral bioavailability? But perhaps not as that is surely what is being determined by the oral (when compared to the IV) studies. The reviewer is lost and would beg clarification.

9. At line 183 the authors are comparing variation on days 1 and day 14. Probably then they are giving within day variation, reducing due to accumulation in the albumin space. But they then immediately go on to talk about 'within patient variability' which could be the same thing (within day) or what is usually understood by clinicians and insulin users, namely between days. The figure they quote for insulin glargine seems to be within-patient, between-day, CV for fasting plasma glucose! The pharmaceutical industry has willingly muddied the water in this area for insulin, and clarity would welcome.

10. In Table 1 relative potency seems to be 50-150 x that of human insulin. Perhaps these are percentages, however, but units are not given.

11. Line 263 et seq: This seems intuitively correct, but again the general reader will have trouble in understanding what is meant by central and peripheral compartments. The confusion partly arises for clinicians because central and peripheral are also used for fat distribution (and much more commonly so), so that 'central' equates to the liver and abdomen. Clarification of meaning is suggested.

12. Incidentally the reviewer also notes that accessibility of albumin bound insulin is different in the liver and the periphery, and wonders (as others will) whether this would be more extreme for these analogues. A similar issue arises with the blood-brain-barrier, possibly explaining some issues relating to insulin detemir. Perhaps the issues could be noted, even if no data are available and speculation as to hepatocyte and hypothalamic access is all that can be given.

13. Figure 5d: Again it is obvious with a little thought which lines are PK and which PD, but the positioning of the keys is not intuitive. They could either be moved to be more closely associated with their respective axes, or even better labelled as PK and PD.

14. in various places in the text rats and dogs are described as the animal model, but not how they were prepared even in summary form. The information might be in the supplementary information, but the animals might at least be described as 'diabetic' or not, and if the former the mode of achieving that and subsequent management when first mentioned.

15. The hypoglycaemia data make sense, but the comparison with human insulin is pretty much meaningless (one intended as a basal insulin, the other a meal-time insulin), as in part the authors note. The only relevant comparison really would be with an infused square wave of human insulin over several hours, which would not be dissimilar to the oral insulin findings perhaps, and thus similar to the findings from Halberg et al. I am therefore not sure the human insulin data presented here helps at all. Incidentally the issue here is not simply relative affects on Ra and Rd, and also ability to switch on Ra in the face of excess insulin after induction of hypoglycaemia due

to prior relative overdosing, or changes in insulin sensitivity. This problem was seen with the Lilly hepato-specific insulin and has not been studied here (I am not suggesting that these insulins in any way are similar to PEG-lispro). A little more caution in discussion and further understanding are required.

16. References: The way journal titles are given varies. Some article titles are first letter capitalized.

17. But reference 32 is a mess, with two spelling howlers, as well as style variation.

Molecular Engineering of Safe and Efficacious Oral Basal Insulin

Dear Reviewers,

We would like to express our appreciation for taking the time and effort to review our manuscript. Attached please find our response to the reviewers' comments.

Yours sincerely,

Thomas Kjeldsen, Dr. Techn.
Senior Principal Scientist
Novo Nordisk A/S
Denmark

Reviewer #1 (Remarks to the Author):

In the communication from Kjeldsen's group, the development of ultra-long acting insulin analogs for the use of oral insulin therapy for diabetic patients has been reported. This is achieved by the combination of the pertinent mutations on the amino acid sequence of human insulin with the modification at the B29 lysine residue with an albumin-binding fatty acid. These modifications indeed increase the life-time of the bioavailability in the plasma significantly. The achievements might have a strong impact in the communities. However, in consideration that one of the oral insulin analogs, i.e., OI338, was already reported and tested for the pharmacokinetic analysis for men in refs 1 and 2, where conceptually the same strategy was employed for the development of oral insulin drugs, scientific advances of the submitted paper would be rather limited. Although several new ex-

perimental data, including new ultra-long acting insulin analogs, such as OI216 and OI320, the crystal structures, and their detailed pharmacokinetic and pharmacodynamic behaviors for dogs, have been presented, the paper is not recommended for publication in Nature Communications due to the lack of enough novelty and innovative data.

Authors response: Although it is true that we have previously published the clinical performance of OI338, the current manuscript describes the years of interdisciplinary scientific research that has been undertaken in order to achieve this successful glucose lowering with oral insulin tablets in man. Therefore, we respectfully disagree with this reviewer's opinion that "...scientific advances of the submitted paper would be rather limited." Quite contrarily, this paper provides the hypotheses and details of the painstaking iterations in molecular design leading to the unprecedented clinical performance of OI338. Indeed, in this manuscript, we focus on a novel molecular and biological concept of designing analogues with extremely low *in vitro* receptor affinity which retain full *in vivo* potency and have the advantage of binding strongly to albumin and distributing very slowly to the peripheral tissues, leading not only to the very long duration of action critical for minimizing variation at steady state, but also providing a cushion/buffering effect towards hypoglycaemia. This is so essential for making oral insulin feasible given the relatively low bioavailability and higher variability inherently associated with oral peptide dosing. Moreover, we present advanced clamp and tracer studies in dogs demonstrating the hypoglycaemia benefits of this type of molecule combined with a fast, peaked PK profile. None of these types of assessments have been conducted in humans and as such these *in vitro* and *in vivo* animal studies strongly complement the human clinical trial results, providing the mechanisms that have led to the successful treatment of type 2 diabetes with oral OI338 tablets. Therefore, we believe that the results described in this manuscript will be of interest to readers of *Nature Communication*.

Reviewer #2 (Remarks to the Author):

In this study, the authors revealed that the combination of changes in three different properties of human insulin considerably increased the feasibility of developing oral insulin. The changes include enhancing the proteolytic stability, increasing the albumin binding affinity and decreasing insulin-receptor binding affinity. In addition, they found that the new engineered molecules may provide direct protection against hypoglycemia, a condition that is usually resulted from insulin overdose and can cause severe health consequences. This research is original and important. The research design is well developed, and the experimental procedures and results are clearly described. The data from this research study support the feasibility of developing oral insulin for diabetes treatment. This work is expected to quickly catch the attention of scientists in the field of insulin. Overall,

this manuscript is within the scope of Nature Communication and I recommend publication as it is.

Authors response: We thank the Reviewer for showing interest in our manuscript and for the favourable comments.

Reviewer #3 (Remarks to the Author):

Nat Com 235956; Hubálek et al

The paper from Hubálek and colleagues et al is an interesting description of an approach to design of insulin analogues for oral administration. The topic of high interest, although whether or not these efforts lead to a marketable product is a subject for future clinical work. The manuscript is dense, but goes to some trouble to explain the rationale of the steps described, and does this with useful clarity.

My comments are generally editorial:

1. The authors vary in their usage of types of abbreviation for amino acids, in a way that is not good communication and can be confusing. At a simple level most clinical readers will not be familiar with the single letter format, and that is not intuitive. In Figure 2a it is confusing, as both notations are used, and 'H' is used for hydrogen and for histidine and for human! The reviewer suggests revising to the 3-letter form for amino acids throughout.

Authors response: We would like to thank the Reviewer for this remark and for the useful suggestions. One-letter abbreviations have been replaced by three letter abbreviations throughout the manuscript wherever possible.

Figure 2a has been adapted according to the reviewer's comments as follows: one-letter abbreviations have been replaced with the three-letter form for amino acids wherever possible. Please note one exception in figure 2 which is the schematic representation of insulin where one-letter abbreviations is used for simplicity and easier comprehension. We hope this is acceptable since in context is clear that this is an amino acids sequence and most readers will have some familiarity with the amino sequence of insulin.

Furthermore, the figure legend to figure 2 has been rewritten to clarify as follows: "Figure 2: Structures of the insulin analogues; note that "R" in HI, OI106, OI189 is hydrogen (i.e. chemically unmodified insulin) and that "R" in OI216, OI338 and OI320 is the chemical modification of B29 (see full drawing of chemical modification); Note that "R" is identical for OI216, OI338 and OI320 as illustrated in 2a)

(all analogues are desB30). Insulin residues that are modified are A14 (Native Tyr, modified Glu), B25 (Native Phe), modified His), B27 Thr, modified by deletion (des), B30 Thr), modified by deletion (des)."

2 Figure 2 is not intuitively laid out in any case. It took the rather slow reviewer some 3 minutes to work out that the three 'R =' usages without anything following referred to the aliphatic side chain displaced further to the right and somewhat lower. Using a brace (curly bracket) with the chemical formula immediately adjacent to it would probably largely resolve the problem.

Authors response: We apologize for the apparent confusion. Figure 2 has been modified as follows. In figure 2 "R= either hydrogen or chemical modification" have been indicated immediately adjacent to HI as well as each insulin analogue. Since the chemical modification is identical for all the modified analogues only one chemical formula is shown. Please also see Authors response 2 to comment # 1 regarding how the figure legend to figure 2 has been rewritten to clarify the usage of "R".

3. In any case what is the significance of the red coloured symbols in this chemical structure? This is not explained in the legend.

Authors response: the red coloured symbols in the chemical structure have been replaced by black symbols.

4. Again while 'Ca atoms' will be known to the authors, the reader unfamiliar with the analogues and size estimates will wonder if calcium is really meant. Suggest spell it out on first usage. Oddly zinc is always spelt out in the supplementary material.

Authors response: Here it is the font that appears to result in the problem. Ca is not Ca, calcium, but C(alpha), coordinates of alpha carbon atoms in the X-ray structures. At the first occurrence, this has been spelled out. "The crystal structures of OI338 and OI320 showed a conserved insulin structure with small differences compared to the human insulin backbone, with root mean square deviations of 0.37Å (OI338, 44 C(alpha), Ca atoms) and 0.56Å (OI320, 41 Ca atoms)"

5. Sodium caprate will not be familiar to many. It is briefly introduced and referenced however. It is not a US FDA GRAS substance I believe. Perhaps it is already used in some other orally administered substances (apart from being present in

some foods). Some statement on its regulatory status in Europe and the US would be welcome.

Authors response: Good point. The following have been inserted in "Methods" section describing "Tablet Formulation": The free acid of sodium caprate – capric acid constitutes close to 10% of coconut oil and up to 1 % in coconut milk and it's generally recognized as safe. As a food additive it is regulated in the US under the Electronic Code of Federal Regulations, e-CFR chapter 172.860 – fatty acids states that it may safely be used in food and in manufacture of food [https://www.ecfr.gov/cgi-bin/text-idx?SID=c7e0678182071d4156396f4bf749a8a7&mc=true&node=se21.3.172_1860&rgn=div8].

6. In the Abstract the authors comment on 'efficient' glucose lowering in animals in clamps. While it is obvious to the authors, some readers may not realize that IV administration is used in these studies. It is suggested to say that.

Authors response: we agree that this was not expressed clearly and, "glucose lowering" is not an endpoint during a glucose clamp. Therefore, this section of the abstract has been rephrased for clarification as follows "These low receptor affinity insulin analogues still stimulated efficient glucose disposal in healthy and conscious rats, pigs and dogs during constant *iv* infusion and euglycemic clamp conditions."

7. It is anyway not clear how the IV dosing relative the human insulin was arrived at. To this reviewer efficiency would imply that, at steady state with albumin binding saturated, the dose giving 100 % equivalent in figure 2 would have the same molar content as the human insulin standard. Was that the case? (Authors response: Yes, that was the case)? Otherwise efficacy is established not efficiency.

Authors response: Legend to figure 2 is rephrased for clarification as follows "*d) Potency of oral insulin analogues relative to human insulin on whole body glucose disposal during different constant molar iv infusion rates to steady state (see method section) under euglycemic clamp in healthy and conscious rats (circles), dogs (triangles) and pigs (squares), (for confidence intervals see table 1)*"

Figure 4 on the horizontal axis is indeed molar (and shows a halving of potency, which is indeed a minor difference), but the authors could be more explicit in the relevant parts of the text as to doses used.

Authors response: Thank you, to address this point the following has been inserted: "At least three different doses (constant molar infusion rates) of human insulin and the oral insulin analogue in question were tested in the dose response glucose clamp studies in rats, dogs and pigs. The doses were chosen individually in the three different species to be at the linear part of the log(dose) response curve based on historical data for human insulin and on *a priori* assumptions for the oral insulin analogues."

This statement is now included the section: **Pharmacodynamic potency determined as whole-body glucose disposal**

8. In Figure 3 the oral doses are described as 'dose normalized'. This might mean just about anything, but perhaps means doses were adjusted for the oral bioavailability? But perhaps not as that is surely what is being determined by the oral (when compared to the IV) studies. The reviewer is lost and would beg clarification.

Authors response: Here the meaning of "Dose normalized" is 'the plotted concentrations corresponds to exposure divided by the oral dose in nmol/kg for each insulin analogue. This information is now included in the figure text for figure 3 as follows: Pharmacokinetic profiles of OI106 (triangles), OI216 (squares) and OI338 (circles) after single oral dosing in healthy and conscious dogs (n=8), the plotted concentrations corresponds to exposure divided with the oral dose in nmol/kg for each insulin analogue.

9. At line 183 the authors are comparing variation on days 1 and day 14. Probably then they are giving within day variation, reducing due to accumulation in the albumin space. But they then immediately go on to talk about 'within patient variability' which could be the same thing (within day) or what is usually understood by clinicians and insulin users, namely between days. The figure they quote for insulin glargine seems to be within-patient, between-day, CV for fasting plasma glucose! The pharmaceutical industry has willingly muddied the water in this area for insulin, and clarity would welcome.

Authors response: We admit that it is difficult to understand what is being discussed in terms of variability, both in our dog study and in reference to the clinical trial data that is referenced. Therefore, for the sake of clarity, we have specifically defined what was measured in the different studies and have described how the variation was determined and the manuscript modified as follows (line 207 to 220).

"A multiple dose study wherein OI338 was dosed daily orally to dogs for 14 days demonstrated that accumulation to a steady state exposure occurs after 5 days. The intra-/inter-individual coefficient of variation, determined as the within day variation observed in the group, was found to be reduced over time as can be seen by less peaked individual PK profiles and less variation between dogs on day 14 (40%) compared to day 1 (100%) (Figure 3d and e). In man, in an eight week study, the variability observed for fasting plasma glucose for the OI388 group was found to be low with a coefficient of variation of 23.2 % after oral administration which is similar to 17.5% for insulin glargine after SC administration¹. General pharmacokinetic parameters in healthy and conscious Beagle dogs are summarised in Table 1."

Furthermore, the following have been added to the legend of figure 3 (line 236-238) "Coefficient of variation of bioavailability based on area under the curves in fig d), multiple dose study. Intra/inter-individual CV based on the mean of the group was reduced from 100 to 40% after treatment to steady state."

10. In Table 1 relative potency seems to be 50-150 x that of human insulin. Perhaps these are percentages, however, but units are not given.

Authors response: Unit (%) now added to table 1

11. Line 263 et seq: This seems intuitively correct, but again the general reader will have trouble in understanding what is meant by central and peripheral compartments. The confusion partly arises for clinicians because central and peripheral are also used for fat distribution (and much more commonly so), so that 'central' equates to the liver and abdomen. Clarification of meaning is suggested.

Authors response: Thank you for pointing out the ambiguity in what we mean by "central" and "peripheral" compartments when referring to the distribution of insulin. We have now introduced specific definitions as well as examples of the two compartments (lines 291-294) as follows:

"We modelled the steady-state distribution between the central (systemic circulation/plasma, including the liver) and peripheral (tissue/interstitial fluid, including metabolically relevant tissues for insulin such as fat and muscle) compartments to assess differences between orally administered OI338 and SC delivered human insulin."

12. Incidentally the reviewer also notes that accessibility of albumin bound insulin is different in the liver and the periphery, and wonders (as others will) whether

this would be more extreme for these analogues. A similar issue arises with the blood-brain-barrier, possibly explaining some issues relating to insulin detemir. Perhaps the issues could be noted, even if no data are available and speculation as to hepatocyte and hypothalamic access is all that can be given.

Authors response: We agree that this is a very good point and we have thus added a paragraph where we have discussed both the physiological liver exposure and possible differences in CNS distribution. Furthermore, reference to PEGlispro (response to point #15 below)

"In addition to the advantages that this type of distribution would have to minimize the risk of hypoglycaemia due to variations in oral absorption (see section below on hypoglycaemia assessment), it could be speculated that an insulin analogue with strong albumin binding having a rapid, peaked central (blood compartment) PK profile would mimic the pattern of insulin secreted from the pancreas directly into the portal circulation where the liver is highly exposed. This peaked PK profile upon oral delivery with greater liver effect is not expected to be the same as what was observed for PEGlispro which demonstrated a constant hepato-selective effect¹⁷. Whether this more physiological profile of rapid and peaked insulin delivery to the liver (part of the blood compartment due to the fenestrated capillaries) would lead to improved metabolic control remains to be determined. Similarly, this type of insulin analogue may be expected to have an altered CNS distribution and it is not known whether this would lead to any differences in central effects compared to regular human insulin as has been suggested for insulin detemir¹⁸"

13. Figure 5d: Again it is obvious with a little thought which lines are PK and which PD, but the positioning of the keys is not intuitive. They could either be moved to be more closely associated with their respective axes, or even better labelled as PK and PD.

Authors response: "PK" and "PD" have been added to the symbols indicating which lines are PK and which PD, respectively.

14. in various places in the text rats and dogs are described as the animal model, but not how they were prepared even in summary form. The information might be in the supplementary information, but the animals might at least be described as 'diabetic' or not, and if the former the mode of achieving that and subsequent management when first mentioned.

Authors response: Also, a good point and to clarify “[species] model” is replaced with “healthy and conscious [species]” throughout the manuscript

15. The hypoglycaemia data make sense, but the comparison with human insulin is pretty much meaningless (one intended as a basal insulin, the other a meal-time insulin), as in part the authors note. The only relevant comparison really would be with an infused square wave of human insulin over several hours, which would not be dissimilar to the oral insulin findings perhaps, and thus similar to the findings from Halberg et al. I am therefore not sure the human insulin data presented here helps at all. Incidentally the issue here is not simply relative effects on R_a and R_d , and also ability to switch on R_a in the face of excess insulin after induction of hypoglycaemia due to prior relative overdosing, or changes in insulin sensitivity. This problem was seen with the Lilly hepato-specific insulin and has not been studied here (I am not suggesting that these insulins in any way are similar to PEG-lispro). A little more caution in discussion and further understanding are required.

Authors response: While it is true that OI320 is a much longer-acting insulin analogue compared to human insulin, a comparison of the metabolic effects *per se* were not in focus, but rather this study was conducted to address the risk of hypoglycaemia that may occur as a result of dosing variations due to possible day to day differences in oral bioavailability. It was important to compare the oral delivery of OI320, representing an acylated insulin analogue that binds to albumin and having a peaked profile that is, during the peak, primarily restricted to the central (blood) compartment to an insulin which quickly distributes to the peripheral compartments like fat and muscle. This difference in distribution can be observed by the relative effects on R_a (liver) and R_d (muscle/fat). We believe that it is important to communicate that the difference in distribution is reflected in a “safer” profile for OI320 compared to oral delivery of a regular insulin which is much more susceptible to hypoglycaemia with changes in absorption due to the rapid effect of R_d . This is valuable information for others who are interested in pursuing the development of oral insulin therapy and highlights the major challenge that is currently being faced with most other oral insulin programs that are based on regular human insulin or fast distributing analogues. We have thus added a discussion of the relevance of the human insulin comparison with regards to hypoglycaemia and caution against the use of human insulin for oral therapy while highlighting the advantages of centrally (blood) restricted insulin analogues such as OI320 (see lines 308-320).

Furthermore, to avoid misunderstandings in the difference in PK exposure in the different compartments, we have focused the modelling shown in supplementary

figure 6 to a comparison of the oral and s.c. profiles of OI338 to s.c. human insulin which is the most relevant.

To reflect this the legend to supplementary figure 6 has been changed as follows:

“The figures show simulations of *SC* and *PO* administration with either human insulin or OI338 in the central (plasma) and peripheral (tissue) compartments, respectively. The simulations are normalized to the area-under-the-curve (AUC) for each plot and are based on data on the insulin formulations tested in dogs. The plots show that *PO* OI338 have an initial phase with a high central-to-peripheral ratio of insulin which *SC* OI338 and *SC* human insulin do not have. Note that “normalised concentration” reflects both free and albumin bound OI338; the majority of OI338 will be associated with albumin.”

16. References: The way journal titles are given varies. Some article titles are first letter capitalized.

Authors response: the reference list has been revised. Thus, it now fulfils the requirement set by *Nature Communications*.

17. But reference 32 is a mess, with two spelling howlers, as well as style variation.

Authors response: Reference 32 have been corrected

Reviewers' Comments:

Reviewer #3:

Remarks to the Author:

No further comments

Reviewers' Comments:

Reviewer #3:

Remarks to the Author:

No further comments

Authors response: we appreciate the review.